# Exploring Black Soybean Extract Cream for Inflammatory Dermatitis—Toward Radiation Dermatitis Relief

**DOI:** 10.3390/ijms252111598

**Published:** 2024-10-29

**Authors:** Hsin-Hua Lee, Yu-Hsiang Huang, Joh-Jong Huang, Ming-Yii Huang

**Affiliations:** 1Ph.D. Program in Environmental and Occupational Medicine, National Health Research Institutes, Kaohsiung Medical University, Kaohsiung 807, Taiwan; hhlee@kmu.edu.tw; 2Department of Radiation Oncology, Kaohsiung Medical University Hospital, Kaohsiung Medical University, Kaohsiung 807, Taiwan; 3Department of Radiation Oncology, Faculty of Medicine, School of Medicine, College of Medicine, Kaohsiung Medical University, Kaohsiung 807, Taiwan; 4Center for Cancer Research, Kaohsiung Medical University, Kaohsiung 807, Taiwan; 5Department of Radiation Oncology & Proton and Radiation Therapy Center, Kaohsiung Chang Gung Memorial Hospital and Chang Gung University, College of Medicine, Kaohsiung 833, Taiwan; j580580@cgmh.org.tw; 6Department of Family Medicine, Kaohsiung Medical University Hospital, Kaohsiung Medical University, Kaohsiung 807, Taiwan; 7Department of Gerontological and Long-Term Care Business, Fooyin University, Kaohsiung 831, Taiwan

**Keywords:** soy, black soybean, extract cream, natural product, *Glycine max* (L.) Merr., dermatitis

## Abstract

We aimed to evaluate the effect of black soybean extract cream (BSEC) on 2,4-dinitrochlorobenzene (DNCB)-induced dermatitis in murine models mimicking inflammatory dermatitis observed in humans. In this DNCB-induced model, BALB/c mice were spread with 100 μL of 2% DNCB twice a week for two weeks to induce skin inflammation on the shaved back skin; then, a placebo or BSEC that consisted of the volatile fraction derived from the seeds of *Glycine max* (L.) Merr. was applied to the DNCB-sensitized mice for 7 days. Gross visual analysis was conducted to assess the impact of BSEC on dermatitis, and an enzyme-linked immunosorbent assay (ELISA) was subsequently performed to detect inflammatory cytokines in the presence or absence of BSEC after DNCB sensitization. Lastly, the possible mechanisms responsible for the effects of BSEC on inflammatory dermatitis were investigated in a human leukemia monocytic cell line, THP-1. Our study showed that BSEC displayed antioxidant and anti-inflammatory effects. BSEC has the ability to diminish dermatitis, and all three experiments demonstrated that BSEC effectively reduced the progression of dermatitis while significantly suppressing inflammatory responses in the preclinical models. Consequently, BSEC exhibited promising phytotherapy for inflammatory dermatitis, potentially attributed to its anti-inflammatory and antioxidant properties.

## 1. Introduction

Dermatitis is an inflammatory cutaneous condition where the disease progression is thought to involve a complex interaction of various inflammatory cytokines, such as interleukin-6 (IL-6) and tumor necrosis factor-alpha (TNF-α), along with the accumulation of oxidative stress [1]. In 1947, soybean lecithin was reported to be a stabilizer of fat, vitamin A, and carotene in foods and feeds [2], and in containing such exceptional phytochemical contents, soybeans have long been used for their antioxidant properties. They can have diverse seed coat colors, varying from yellow, black, brown, and green to bicolor [3]. The crop contains a broad array of ingredients such as oligosaccharides, amino acids, and isoflavone glycosides while possessing immune-regulatory and analgesic functions. Black soybeans [*Glycine max* (L.) Merr.] exhibit various effects, including anti-inflammation and detoxification [4,5], yet are not as prevalent as yellow soybeans that are mainly used for manufacturing soymilk, tofu, and other soy products in the food industry.

Researchers have found that antioxidant activity is higher in black soybeans while possessing isoflavones, such as glycitin and glycitein, that are not present in yellow soybeans [6]. Raffinose, a member of oligosaccharides, is rich in soybeans and is reported to modulate skin differentiation [7]; genistein, a phytoestrogen in soybeans, could suppress inflammation and relieve neuropathic pain in diabetic mice via subcutaneous injection [8]; while anthocyanins extracted from black soybean seed coats have been shown to exert anti-inflammatory effects on keratinocytes, ischemia–reperfusion injury in rat skin flaps, and human gastric epithelial cells [9,10]. While soybean products are generally regarded as safe, a few case reports have shown that malleated soybean oil has been linked to contact dermatitis and categorized as a cosmetic allergen [11].

To date, studies have produced varying outcomes, ranging from potential anti-inflammatory advantages for the skin to instances of allergic contact dermatitis [11,12]. We sought to determine whether the effect of the application of black soybean extract cream (BSEC) could diminish inflammatory dermatitis. We hypothesized that BSEC might be theoretically effective in treating inflammatory dermatitis based on the antioxidant activity of black soybean by inhibiting reactive oxygen species (ROS) generation and subsequent MAPK signaling [13].

## 2. Results

Figure 1A is the schematic diagram of the study design. In order to investigate the effects of BSEC on dermatitis, the DNCB-induced dermatitis model was employed. The skins of mice were sensitized with DNCB twice a week for two weeks and then treated with BSEC every day from days 0 to 6. All mice developed skin hypersensitivity reactions, such as severe hyperkeratosis and thick epidermis, after the DNCB sensitization.

These symptoms were relieved after BSEC treatment (Figure 1B). The negative control group was treated with acetone and olive oil at a 4:1 ratio (the solvent of DNCB). From the images in Figure 1, it is clear that the skin of mice in the negative control group did not exhibit any inflammatory reactions, indicating that the inflammation was caused by DNCB and not by acetone. In Figure 1C, the dermatitis scoring of the skin lesion revealed a gradual decrease from 87% on day 0 to 33% on day 6 in the DNCB group. Additionally, daily applications of BSEC on the skin of DNCB-sensitized mice dramatically reduced the dermatitis severity from 81% on day 0 to 6% on day 6 (Figure 1C). When comparing the DNCB group to the BSEC-treated DNCB group, significant differences were observed from days 4 to 6 (*p* ˂ 0.05), indicating that BSEC treatment could neutralize dermatitis symptoms such as itchiness, lesion development, and lesion enlargement in DNCB-sensitized mice.

Furthermore, we examined the level of inflammatory cytokines, including IL-6 and TNF-α, in the DNCB-induced dermatitis animal model. Figure 2A is the schematic diagram of the study design. The skins of the mice were sensitized with DNCB every 5 days for five times starting from day 0. Mice in the DNCB + BSEC group were treated with BSEC daily from day 0 to day 25. ELISA results showed that DNCB-mediated skin sensitization dramatically increased IL-6 levels in the skin tissues compared to the negative control group (Figure 2B); however, the increase in IL-6 was significantly suppressed by BSEC treatment. TNF-α results showed a similar trend as well (Figure 2C).

Similar effects were observed in vitro in a human leukemia monocytic cell line, THP-1. Black soybean extract (BSE) significantly reduced the lipopolysaccharide (LPS)-induced release of IL-6 (Figure 3A) and TNF-α (Figure 3B) from PMA-differentiated THP-1 macrophages.

## 3. Discussion

Numerous studies have demonstrated the beneficial effects of soybean varieties on different diseases [14,15], as the antioxidant properties in soy products neutralize ROS [16]. Chronic inflammatory responses were thought to be a major cause of dermatitis; indeed, a medicine derived from fermented black soybean [*Glycine max* (L.) Merr.] has been shown to lighten atopic dermatitis via decreasing PKC and IL-4 in mice [17], with the authors finding that the decreased PKC and IL-4 were associated with the down-regulation of inflammatory-related cytokines, such as interleukins and TNF-α. BSEC in our present study also showed a similar downward trend of pro-inflammatory markers in the DNCB-induced dermatitis mice, in which increased IL-6 and TNF-α in the DNCB group were suppressed by BSEC. The mechanism of such action lies in the main ingredient of the seed extract of *Glycine max* (L.), Raffinose, and Stachyose. They can be used as a hydrating moisturizing agent because of many -OH groups (Appendix A), which easily form hydrogen bonds with water [18,19]. Furthermore, through the cream formulation, a thin film develops on the skin that physically slows down the evaporation of water from the epidermis to achieve a moisturizing effect [20]. While some studies focused on black soybeans and skin inflammation [4,9,13,21], to the best of our knowledge, the present study is the first to utilize black soybeans in cream form to counteract dermatitis.

The latest recommendations of the American Academy of Dermatology establish the use of natural products as prevention and the first therapeutic step for dermatitis [22]. Plant-based products avoid synthetic chemicals and artificial additives, reducing possible skin irritation and adverse reactions. Additionally, plant-based products are more sustainable, sourced from renewable resources and produced using eco-friendly methods, thus lowering their carbon footprint. Our study showed that BSEC displayed anti-inflammatory effects. In our previous studies, extracts of black soybean seeds were found to relieve acute radiation-induced dermatitis (RID) in Sprague Dawley rats [23]. Other investigators analyzed a total of 58 human clinical trials of the antioxidant effect of soybeans, indicating the potential of soy in redox processes [16]. Given the proximity between the skin and the radiotherapy (RT) target, RID is observed mostly in breast and head-and-neck cancer patients. As of today, there remains no single treatment other than corticosteroids that efficiently manages RID [24,25]. RT is known to trigger the release of cytokines, including IL-1, IL-6, IL-8, and TNF-α [26]. A portion of patients under RT experienced RID, and severe reactions occurred in 10~25% of head-and-neck cancer patients [27]. Several topical management compounds such as curcumin, hyaluronate, hydrocortisone cream and dressing, aloe vera, chamomile, Dead Sea products, olive oil, and turmeric wood oil have been proposed for RID in clinical trials [27,28,29,30,31,32,33,34,35,36,37]. The application of topical corticosteroids influenced the incidence of wet desquamation and the intensity of RID in female breast cancer patients [37]; however, long-term use of topical corticosteroids increases the risk of skin atrophy [38]. Generally, these agents can be classified as anti-inflammatory, antioxidant, or antibacterial based on their mechanism of action. To investigate the effects of BSEC on skin inflammation, a DNCB-induced dermatitis animal model was employed, where mice were sensitized using DNCB to induce skin inflammation. Dermatitis caused by DNCB and RID both exhibit similar pathological features (ulceration, dermal thickening, inflammation, follicular dropout, sebaceous gland dropout, etc.) and molecular expressions (increased inflammatory factors IL-6 and TNFα). Murine contact hypersensitivity is a frequently used animal model of human allergic contact dermatitis [39], and studies have shown that soybean consumption down-regulated the gene and protein expression, thereby affording protection against dermatitis in mice. Additionally, soymilk consumption may be of therapeutic value for patients with allergic contact dermatitis [40]. The consumption of soy isoflavones reduced contact hypersensitivity symptoms and suggested that the gut microbiota influenced their suppressive activities on dermatitis [41]. Therefore, it is hoped that topical BSEC treatment for DNCB-induced dermatitis with similar pathological features and mechanisms can be extrapolated to treat RID, which is a complication frequently encountered, although still without the most feasible elixir.

Dorjsembe et al. utilized two black soybean cultivars, A63 and Seritae (ST), grown in Paju, Korea, adhering to standard agricultural practices [21]. After air drying and storage, soybeans were processed into a fine powder and extracted with ethanol. Isoflavones and flavonoids were analyzed using HPLC and mass spectrometry, respectively. For the animal study, BALB/c mice were treated with soybean extracts, and skin inflammation was induced. Immunohistochemistry and RT-qPCR analyzed tissue samples, while fibroblast cell cultures underwent Western blot analysis. Statistical significance was determined using ANOVA followed by Tukey’s test. A63 extract demonstrated efficacy in reversing atopic symptoms and inhibiting AD markers, suggesting its potential clinical relevance. This study focused on atopic dermatitis in an oxazolone-induced murine model. In the present study, we aimed to probe into a promising new approach for improving dermatitis and represent the inaugural exploration of BSEC’s potential in this application.

In various experimental models, genistein, the main isoflavone found in soy, has demonstrated both antioxidative and neuroprotective properties [42]. In the present study, the vapor fraction of whole black soybeans, known as BSEC, was utilized to preserve the complete nutritional content of the black soybeans. As in the case of the soy foods previously evaluated, there does not seem to be a clear protective effect of isolated isoflavones on oxidative stress [16]; although, in the present study, we measured pro-inflammation markers, specifically TNF-α and IL-6, to represent the anti-inflammatory effects of BSEC. Elevated TNF-α and IL-6 levels have been found in patients with severe radiation-induced oral mucositis as well [43]. 

Su et al. also reported an in vivo assay showing that 18β-glycyrrhetinic acid eased RID, reduced inflammatory cell infiltration, and decreased TNF-α and IL-6 levels in cutaneous tissues [44]. They proposed that this favorable effect was probably via the inhibition of NADPH oxidase activity, ROS production, the activation of p38MAPK and nuclear factor-kappa B (NF-κB) signaling, and the DNA-binding activities of NF-κB and AP-1 [44]. NF-κB transcription factor plays a critical role in regulating radiation-induced inflammatory and immune responses [45]. Although further research is needed, the available human clinical trials suggest promising results in utilizing soybean products for their redox effects [16]. Black soybean extract significantly reduced the LPS-induced release of IL-6 and TNF-α from PMA-differentiated THP-1 macrophages, as shown in Figure 3. A recent study on the gene expression response to in vitro challenge with increasing concentrations of LPS reported a similar result between spirulina-fed and soy-fed animals [46]. Despite the observed differences in β-carotene and phenol plasma concentrations in spirulina, compared to soybean-fed dairy cows and fattened bulls, respectively, the expression of antioxidant enzymes hardly differed, indicating a similar high antioxidant power of soybeans.

In this study, daily applications of BSEC on the skin of DNCB-sensitized mice dramatically reduced dermatitis severity from 81% on day 0 to 6% on day 6 (Figure 1C). For the regulation of botanical ingredients in cosmeceuticals, the Expert Panel for Cosmetic Ingredient Safety concluded that soy proteins and peptides are safe in cosmetics under present usage practices [47]. BSEC reduced scratch numbers and dermatitis lesion length, and compared to the placebo, the degrees of erosion and ulceration of the development of punctuate or coalescing crusts were much less. Topical therapeutic compounds with soy products have been used in clinical practice to counteract cutaneous dryness, atrophy, fine wrinkling, and poor wound healing for postmenopausal women [48]. The limitation of the present study was firstly the absence of histopathological analysis due to the specific euthanasia method employed—deep anesthesia with isoflurane followed by cardiac perfusion with saline. This approach significantly compromised tissue integrity, thereby affecting the quality and suitability of the tissues for histopathological examination. We are committed to maintaining the highest ethical standards in our research, and this method was chosen to minimize animal suffering. Additionally, the vehicle cream without black soybean extract has not been tested as a control.

In future studies, we will elucidate the ROS assessment and focus on comprehensive safety assessments so that the results of this study will translate to clinical practice. The severity of dermatitis can be evaluated using the National Cancer Institute—Common Terminology Criteria for Adverse Events (NCI-CTCAE) in future clinical trials. Further extension of the monitoring period for assessing any long-term effects of BSEC would be preferred in future studies, while more detailed analysis for a better understanding of BSEC’s therapeutic potential and the underlying mechanisms would be desirable.

## 4. Materials and Methods

### 4.1. Study Medication and the Extraction Method

Black soybean extract (coordinates of plant picking: 23.25632° N, 120.27372° E) in liquid form (BSEL; liquid form) and BSEC, a cream-based product, comprise the vapor fraction of the seeds of *Glycine max* (L.) Merr., in which the main active ingredients include Raffinose pentahydrate and Stachyose hydrate (Raffinose: 0.0063% and Stachyose: 0.0204%). Ingredients of formulation include the seed of *Glycine max* (L.) Merr. extract and aqua, stearic acid, stearyl alcohol, Tween 80, benzyl alcohol, borneol, and potassium hydroxide. The structures of active compounds are included as Appendix A. 

BSEL and BSEC (Batch No. FP020-14110501) were purchased from the manufacturer, Charsire Biotechnology Corporation, Tainan City 744, Taiwan (Tainan Science Park). The extraction method was described as follows. Seed extracts of *Glycine max* (L.) Merr. were ground into power, and 70% by weight of ethanol or distilled water was applied as an extracting solution; the ratio (*w*/*v*) of the seeds of *Glycine max* (L.) Merr. and the extracting solution was about 1:10. The seeds of *Glycine max* (L.) Merr. were extracted at a barometric pressure of about 1 atm at a temperature of about 45 °C to obtain a crude extract. The solids were removed from the crude extract to obtain a liquid portion. The liquid portion was further concentrated by a reduced-pressure condenser to obtain a concentrated solid portion. The concentrated solid portion was further dried at 70 °C. The seeds of *Glycine max* (L.) Merr. were minded into power, and 2% by weight of ethanol or distilled water was applied as a second extracting solution; the ratio (*w*/*v*) of the seeds of *Glycine max* (L.) Merr. and the second extracting solution was about 1:10. The vapor fraction was obtained by vaporizing the soybean seeds in a rotary evaporator (EYELA N-1000S, 1000S-W) at a pressure of lower than 1 atm and a temperature of 90 °C, passing through a condensing tube supplied with cold water.

Prior to the present study, both study materials were examined in a 13-week study of dermal administration in minipigs with a 14-day recovery period in Charles River Laboratories (Spencerville, OH 45887, USA) that confirmed safety use. The report was ordered by the manufacturer of BSEL and BSEC (Charsire Biotechnology Corp. in Tainan City 744, Taiwan (Tainan Science Park)). The dermal administration of CSTC1 once daily was well tolerated in minipigs at levels of 0.76, 2.28, and 3.8 mg/kg/day. The no-observed-adverse-effect level (NOAEL) was determined to be 3.8 mg of the cream’s active ingredient per kg per day. Considering the cream contains 0.1% of the active ingredient (CSTC1), and the dosage conversion factor between mice and pigs is 9, the dosage was selected to be at the dose of 0.5 g BSEC per mouse per day according to the NOAEL and our previous study [23]. The dosage used in mice falls within the toxicological safety range established for mini pigs since 0.5 g of cream per day in a 30 g mouse is equivalent to a dosage of 1.85mg of the active ingredient per kg per day in mini pigs.

### 4.2. DNCB-Induced Dermatitis Animal Model and Dermatitis Severity Scoring

Our research was conducted in accordance with internationally accepted principles for laboratory animal use and care. These animal studies were reviewed and approved by the Institutional Animal Care and Use Committee (IACUC) of Kaohsiung Medical University (IACUC Approval No. 102043). Six-week-old male pathogen-free BALB/c mice were purchased from the National Laboratory Animal Center (Taipei City, Taiwan) and maintained under controlled temperature (25 ± 1 °C) and humidity (60 ± 5%) with a 12 h light/12 h dark cycle. Three mice were housed together per cage with free access to water and chow and were then divided into three groups as follows: negative control group (N = 3), DNCB group (N = 6), and DNCB group with BSEC treatment (N = 6). The images in Figure 1 show that the skin of the negative control group remained in a normal state, clearly different from the pathological changes observed in the DNCB group. Hence, three mice were sufficient to represent the negative control. Additionally, this approach aligns with ethical considerations to reduce animal usage. For dermatitis induction, all mice had 100 μL of 2% DNCB applied on the hair-shaved back twice a week for two weeks prior to BSEC treatment. BSEC was then applied to the sensitized skin at the dose of 0.5 g every day from day 0 to day 6. The negative control group was treated with acetone and olive oil at a 4:1 ratio (the solvent of DNCB).

The severity of dermatitis was evaluated by a modified dermatitis scoring system [49], as shown in Table 1, developed as a scale ranging from zero (normal) to 100 (most severe) based on three aspects, including (A) scratching number, (B) character of lesions, and (C) length of skin lesions. Scores were calculated from days 0 to 6 after DNCB application and applied to the formula [(A + B + C)/9] × 100 to determine the severity of dermatitis. All the experimental procedures were performed under the Guideline for Animal Experiments of the Animal Center at Kaohsiung Medical University in Kaohsiung, Taiwan (IACUC Approval No. 102043). 

### 4.3. Enzyme-Linked Immunosorbent Assay (ELISA)

#### 4.3.1. Cell Culture and ELISA

THP-1 cells were purchased from Bioresource Collection and Research Center in Taipei City, Taiwan, and cultured in RPMI1640 supplemented with 10% FBS at 5% CO_2_ and a 37 °C incubator. After the cell density reached 1 × 10^5^ to 1 × 10^6^ cells/mL, the cells were then seeded at 1 × 10^6^/well (3.5 cm in diameter) containing 75 µM phorbol 12-myristate 13-acetate (PMA) to induce monocyte–macrophage differentiation for 42 h. PMA-differentiated THP-1 macrophages were pre-treated with 0.175, 0.262, or 0.349 mg/mL BSEL or 40 µM JSH-23 (N = 3/group) 1 h prior to stimulation with 1 µg/mL LPS (from *E. coli* O55:B5) for another 6 h. JSH-23 is an inflammation inhibitor. It can inhibit NF-κB transcriptional activity. The control group and BSEL (0.262 mg/mL)-only group were not stimulated with LPS. The supernatants of each testing group were collected and preserved at −80 °C. The levels of IL-6 and TNF-α were assessed using an enzyme-linked immunosorbent assay (ELISA) system (Invitrogen, Vienna, Austria) according to the manufacturer’s instructions. Absorbance at 450 nm was determined by a microplate spectrophotometer (BMGLABTECH—SPECTROstar Nano, 77799, Ortenberg, Germany).

#### 4.3.2. Mice and ELISA

For ELISA, mice were assigned into three groups (N = 8 in each group): negative control, DNCB, and DNCB + BSEC groups. The rationale was to compare the effects of BSEC treatment post-induction versus during induction of dermatitis. A shorter induction interval (twice a week) might influence the treatment outcome; hence, it was modified to once every 5 days. For dermatitis induction, mice with a shaved back had 100 μL of 2% DNCB applied every 5 days for 20 days. BSEC was then applied to the sensitized skin at the dose of 0.5 g every day after the first DNCB application. The negative control group was treated with acetone and olive oil at a 4:1 ratio (the solvent of DNCB). Ultimately, skin samples were collected from the mice on day 25 after the initial DNCB application, with the total protein of the skin being extracted by ultrasonic homogenizer in a radioimmunoprecipitation assay (RIPA) buffer with protease inhibitors. After centrifugation, the supernatant was collected and preserved at −80 °C, and then concentrations of cytokines (IL-6 and TNF-α) in the skin were assessed by ELISA kits (Thermo Fisher Scientific, Inc., Waltham, MA, USA) following the manufacturer’s instructions.

### 4.4. Statistical Analysis

Statistical analysis was performed using the Statistical Analysis Software version 9.4 (SAS) program (SAS Institute, Cary, NC, USA), and the results were presented as mean ± standard error of the mean (S.E.M) or standard deviation (SD) as indicated. Statistical *p*-values were determined by Student’s unpaired *t*-test and considered statistically significant if *p*-value < 0.05.

### 4.5. ARRIVE (Animal Research: Reporting of In Vivo Experiments) Checklist Compliance

Criteria for Euthanasia: Mice were euthanized when they exhibited a weight loss of 20% or were too weak to eat or drink for 24 h.

Duration of the Experiment: The experimental period lasted a maximum of 25 days.

Animal Numbers and Outcomes: A total of 39 mice were used in the experiment. There were no accidental deaths during this study, and all animals were euthanized at the end of the scheduled experimental period.

Monitoring Frequency: The health and behavior of the mice were monitored daily.

Animal Welfare Considerations:

The study protocol was approved by the Institutional Animal Care and Use Committee (IACUC) and adhered to the principles of the 3Rs (Replacement, Reduction, and Refinement).

Personnel involved in animal handling were required to undergo educational training and certification to minimize animal suffering and unnecessary sacrifice.

Mice were provided with adequate food, water, and a clean environment.

Mice were housed in groups with environmental enrichment.

Daily health checks were conducted, and appropriate treatments were administered as needed.

Anesthetic Administration and Dosage:

The route of administration for induction and therapeutic purposes was a topical application on the dorsal skin.

The mice were induced with anesthesia using 4% isoflurane in a pre-anesthesia chamber. Subsequently, anesthesia was maintained at a concentration of 2% isoflurane on the surgical platform.

Method of Euthanasia: Mice were euthanized by deep anesthesia with isoflurane followed by cardiac perfusion with saline.

Verification of Death: Death was confirmed by the complete cessation of heartbeat and respiration and the dilation of pupils.

## 5. Conclusions

The data presented in this study showed that the application of BSEC reduced the severity of dermatitis and suppressed inflammatory responses in the murine model of DNCB-induced dermatitis. BSEC exhibited aesthetic benefits of natural ingredients via significant inhibition of skin inflammation, leading to a decrease in the expression of pro-inflammatory mediators. These findings suggest that BSEC holds therapeutic potential for mitigating inflammatory dermatitis and could be considered as a complementary and alternative medicine.

## Figures and Tables

**Figure 1 ijms-25-11598-f001:**
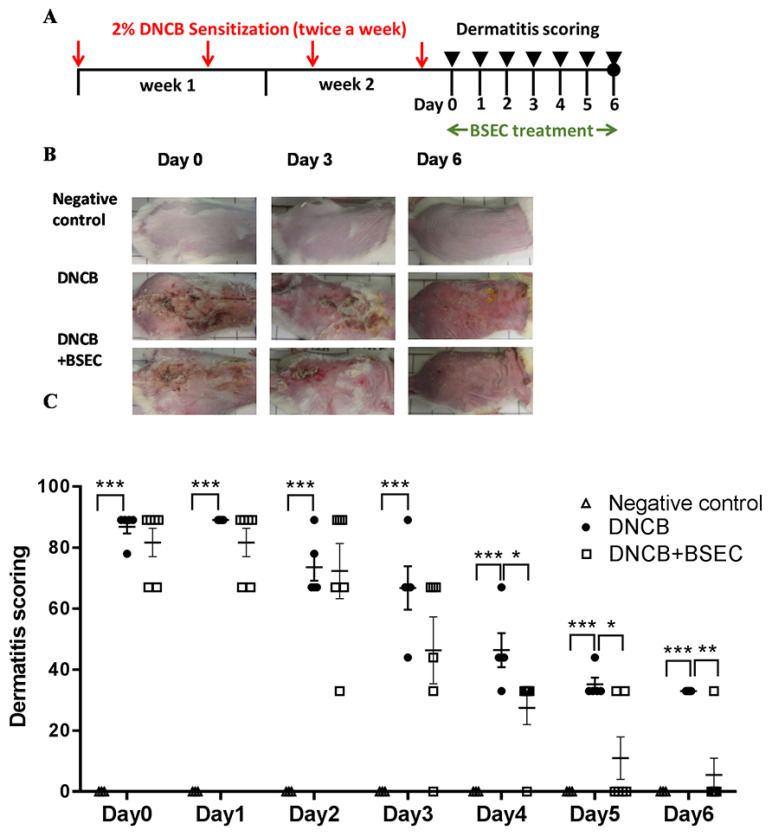
BSEC accelerated the recovery of dermatitis induced by DNCB in vivo. (**A**) Schematic diagram of the study design. (**B**) Representative photos of negative control mice and mice treated with DNCB or DNCB plus BSEC. (**C**) The dermatitis scoring was determined from days 0 to 6 after DNCB induction based on the scratch number, character of lesions, and lesion length. Dermatitis scores of individual mice were presented (3–6 mice per group). Mean ± S.E.M were also presented. * *p* < 0.05; ** *p* < 0.01; *** *p* < 0.001 by Student’s unpaired *t*-test. Abbreviations: BSEC, black soybean exact cream; DNCB, 2,4-dinitrochlorobenzene.

**Figure 2 ijms-25-11598-f002:**
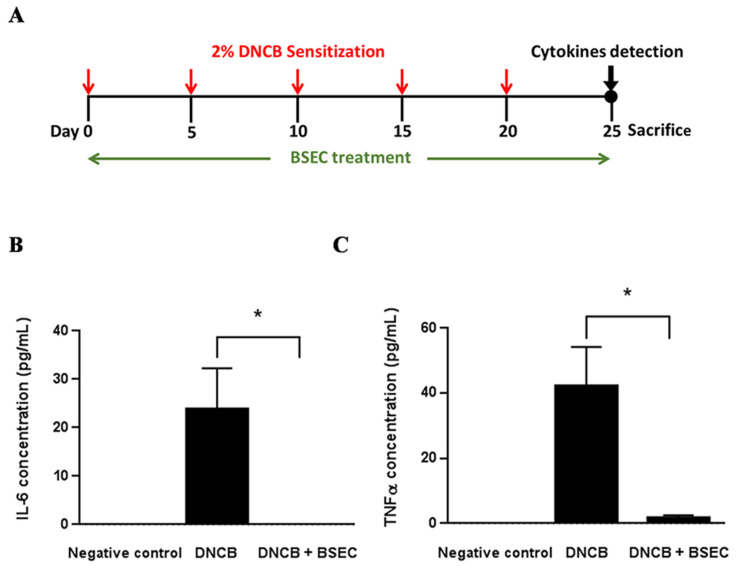
BSEC significantly repressed DNCB-elicited inflammatory responses in vivo. (**A**) Schematic diagram of the study design. Effects of BSEC on (**B**) IL-6 and (**C**) TNF-α level in the skin of DNCB-induced dermatitis mice. Values were presented as mean ± S.E.M with 8 mice per group. * *p* < 0.05 when compared with the DNCB-treated group by Student’s unpaired *t*-test. Abbreviations: BSEC, black soybean exact cream; DNCB, 2,4-dinitrochlorobenzene.

**Figure 3 ijms-25-11598-f003:**
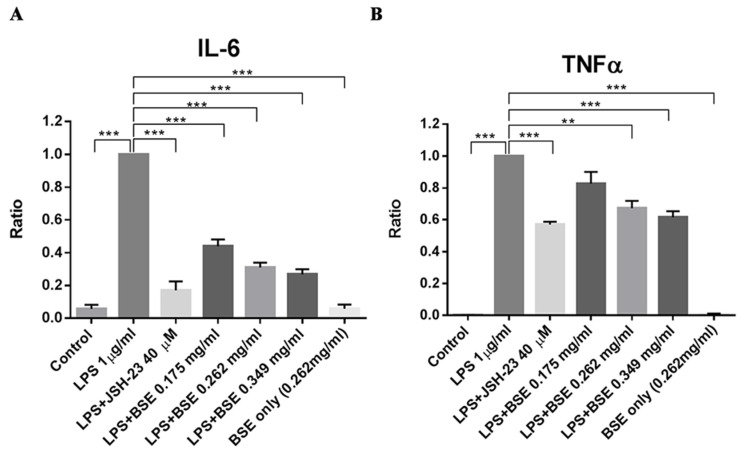
BSEL significantly repressed LPS-elicited cytokine release in vitro, (**A**) IL-6, and (**B**) TNF-α from PMA-differentiated THP-1 macrophages. JSH-23 is an inflammation inhibitor. It inhibits NF-κB transcriptional activity. To obtain the ratio, the concentration (pg/mL) of each cytokine was divided by the concentration of the same cytokine in the LPS group. Ratios were presented as mean ± S.E.M. ** *p* < 0.01; *** *p* < 0.001 when compared with the LPS-treated group by Student’s unpaired *t*-test. Abbreviations: BSE, black soybean exact; LPS, lipopolysaccharide.

**Table 1 ijms-25-11598-t001:** Dermatitis scoring system.

Score	A. Scratch Number	B. Characteristics of Lesions	C. Lesion Length
0	None	No lesion present	0 cm
1	<5	Excoriation only one, small punctuate crust	<1 cm
2	5–10	Multiple, small punctuate crusts or coalescing crust	1–2 cm
3	>10	Erosion or ulceration	>2 cm
Calculated score: [(A + B + C)/9] × 100			

## Data Availability

The data analyzed during the current study are available from the corresponding author upon reasonable request.

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
