# Peer review of "Exploring Black Soybean Extract Cream for Inflammatory Dermatitis—Toward Radiation Dermatitis Relief"

_ijms, 2024, doi:10.3390/ijms252111598_

Round 1

Reviewer 1 Report (Previous Reviewer 1)

Comments and Suggestions for Authors

The authors revised the manuscript properly.

The research finally aimed to apply the Black soybean extract cream (BSEC) to RID. To approach this challenge, in this manuscript, the authors first assessed the inhibition of inflammatory dermatitis by DNCB. Successfully, BSEC suppressed the DNCB-induced inflammation and dermatitis in vivo and corresponding in vitro situations. In the discussion, the authors extended this data to RID as the rational discussion. It opens the possibility of BSEC for treating RID  patients. The data are thoughtful for the potential readers of IJMS and patients suffering from dermatitis.

Author Response

Dear Editors and Reviewer 1,

Thank you for your careful inspection on our manuscript.

Reviewer 1 Comments and Suggestions for Authors

The authors revised the manuscript properly.

The research finally aimed to apply the Black soybean extract cream (BSEC) to RID. To approach this challenge, in this manuscript, the authors first assessed the inhibition of inflammatory dermatitis by DNCB. Successfully, BSEC suppressed the DNCB-induced inflammation and dermatitis in vivo and corresponding in vitro situations. In the discussion, the authors extended this data to RID as the rational discussion. It opens the possibility of BSEC for treating RID patients. The data are thoughtful for the potential readers of IJMS and patients suffering from dermatitis.

We truly appreciate your time and effort reviewing our manuscript.

Sincerely,

Hsin-Hua Lee, MD

Department of Radiation Oncology, Kaohsiung Medical University Hospital, No. 100, Tzyou 1st Road, Kaohsiung 807, Taiwan.

Tel: +886 7 3121101       Fax: +886 7 3118894    E-mail: hhlee@kmu.edu.tw

Reviewer 2 Report (New Reviewer)

Comments and Suggestions for Authors

The results of the BSEC investigations support the potential of natural products in the treatment of inflammatory skin disorders and are in line with earlier dermatology research. What are the most notable differences between these studies and previous findings in dermatology?

Please make the following changes

- at line 87-89, 296-299 it is other font

 - reference 43 has a mistake  

- line 118 is that correct "concentration (pg/mL)"?

Author Response

Dear Reviewer 2,

Thank you for taking the time to review our manuscript. We have carefully considered your valuable feedback and made revisions accordingly. Below, we provide a point-by-point response to your comments, and we have highlighted the changes in the manuscript to ensure they are easily identifiable.

Reviewer 2
The results of the BSEC investigations support the potential of natural products in the treatment of inflammatory skin disorders and are in line with earlier dermatology research. What are the most notable differences between these studies and previous findings in dermatology?

Response: Line 142-144 3. Discussion While some studies focused on the black soybeans and skin inflammation, to the best of our knowledge, the present study is the first to utilize black soybeans in cream form to counteract dermatitis.

Please make the following changes

- at line 87-89, 296-299 it is other font
Response: We have changed all words to the font of Palatino Linotype. Thank you for your careful inspection.

 - reference 43 has a mistake  
Response: Thank you. We have checked that all references are in the correct format.

We have removed “soy” Keller, M.; Manzocchi, E.; Rentsch, D.; Lugarà, R.; Giller, K. Antioxidant and Inflammatory Gene Expression Profiles of Bovine Peripheral Blood Mononuclear Cells in Response to Arthrospira platensis before and after LPS Challenge. Antioxidants 2021, 10, 814

- line 118 is that correct "concentration (pg/mL)"?
Response: Yes, the concentration unit is correct.

We appreciate your diligence and thoughtful suggestions, which have helped improve our work. We hope you find the revised manuscript satisfactory.

Sincerely yours,

Hsin-Hua Lee, MD

Department of Radiation Oncology, Kaohsiung Medical University Hospital, No. 100, Tzyou 1st Road, Kaohsiung 807, Taiwan.

Tel: +886 7 3121101       Fax: +886 7 3118894    E-mail: hhlee@kmu.edu.tw

This manuscript is a resubmission of an earlier submission. The following is a list of the peer review reports and author responses from that submission.

Round 1

Reviewer 1 Report

Comments and Suggestions for Authors

Abstracting

In this paper, the authors assessed the Black Soybean Extract Cream (BSEC) against 2,4-dinitrochlorobenzene (DNCB)-induced dermatitis in mice. BSEC also suppressed the inflammatory response. To investigate it in vitro, the authors used liquid type BSE (BSEL), and showed the corresponding results. The data was properly collected. However, the manuscript is not suitable for publishing because it includes fatal point.

1. Major points

1.1. Purpose and Conclusion Alignment

The title and introduction section includes “Radiation Dermatitis Relief”, and concluded that “BSEC exhibited promising phytotherapy for RID, potentially attributed to its anti-inflammatory and antioxidant properties”(Abstract), “this study is significant being the first to explore the use of BSEC as a viable solution for countering RID” (Conclusion). However, the authors did not use radiation as the inducer of dermatitis. Instead, the authors used 2,4-dinitrochlorobenzene (DNCB), usually used for the assessment of atopic dermatitis. It is true that the author declared this point in the limitation paragraph (Lines 210 - 213), it could not excuse this defect.

1.2. Proper Introduction

If the authors insist the BSEC effectively improved RID, the methodology of DNCB should be properly introduced. The authors should show generally used methodology of assessment for RID.

1.3. Figure Readability

Fine.

1.4. Figure Reliability

1.        In Figure 3, the explanation of JSH-23 should be written in the main text and the legend.

2.        Please unify using BSE (liquid form) or BSEL (M&M section).

1.5. Missing Figures and Unreferenced Figures

None.

1.6. Sufficient Explanation of Results

1.        Control missing: What is the formulation of the cream itself. The vehicle cream which does not contain BSEC group should be assessed for recovery of the dermatitis and anti-inflammatory experiment. If not, this topic should be included in the limitation paragraph.

2.        The authors showed the BSEC contains at least Raffinose and Stachyose, although they are rich, the BSEC may contain other ingredients. So, it is an overstatement as if the anti-inflammatory effects of BSEC are from these sugars. Therefore, the related description should be removed (Abstract) and weakened (Line 162).

1.7. Issues with Statistical Analyses

None.

1.8. Proper Discussion

Fine except the related points of above-mentioned.

1.9. Materials and Methods

1.        In 4.1,Line 243, °N and °E should be inserted.

2.        In 4.3.1. Line 333, Please re-check whether the concentration of PMA (75M) is true. It may be too high.

2. Minor Points

2.1. Typos

Line 189 remove period in et. .

2.2 Grammers

Fine.

3. Reference check

3.1. Percentage of recent works (after 2020)

Fine. Seventeen out of 50 references (34%) are from after 2000.This meets the standard of citing recent studies (over 30%).

3.2. Proper referencing

Fine

Comments on the Quality of English Language

Fine.

Author Response

We appreciated your diligence and made these adjustments accordingly.We hope you find the revised manuscript to your satisfaction.

Sincerely,

Hsin-Hua Lee, MD

Department of Radiation Oncology, Kaohsiung Medical University Hospital, No. 100, Tzyou 1st Road, Kaohsiung 807, Taiwan.

Tel: +886 7 3121101       Fax: +886 7 3118894    E-mail: hhlee@kmu.edu.tw

Reviewer 2 Report

Comments and Suggestions for Authors

The paper presented for review entitled: Exploring Black Soybean Extract Cream on Radiation Dermatitis Relief is a well-written paper that meets the requirements of a research paper. The paper is interesting, but I have a few questions: 1) why was the experiment conducted on mice and not rats? 2) acetone was used as a solvent for DNCB? how do we know whether the inflammation that occurred on the skin was the result of using DNCB and not additionally acetone? 3) why did the negative control consist of only 3 individuals and the other groups of 6?

Author Response

(The authors gave the same response as above.)

Reviewer 3 Report

Comments and Suggestions for Authors

·         Title: The title accurately describes the article’s subject.

·         Abstract: The abstract effectively summarizes the study's objectives, experimental model, and key results. However, providing more details about the methods and the clinical relevance of the findings would enhance the reader's understanding.

·         Introduction: The introduction offers a comprehensive overview of the research problem, with solid references to black soybean properties and the inflammatory mechanisms involved in radiation-induced dermatitis (RID). While the discussion of antioxidant effects is strong, more context could be provided regarding current clinical challenges in managing RID.

·         Materials and Methods: The methods section is detailed and clearly explains the experimental design. The methods are otherwise well-described, but the inclusion of more graphical data to support the text could improve reader comprehension.

·         Conclusion: The conclusions adequately summarize the findings and the potential therapeutic relevance of black soybean extract cream (BSEC) in treating RID. However, the discussion on limitations could be expanded to further highlight the need for additional studies and long-term monitoring.

-Please adjust the formatting of the text and bibliography according to the guidelines of the paper.

-Please adjust the formatting of the bibliography in the text according to the journal guidelines.

-Please improve the quality of the image Supplemental Figure S1, it as it appears to be cropped.

-Please t-student test p goes in lower case and italics.

-Please adjust the formatting of Table 1 according to the journal guidelines.

Author Response

(The authors gave the same response as above.)

Round 2

Reviewer 1 Report

Comments and Suggestions for Authors

Major points

  1. The authors have adequately addressed most of the points raised, but the most critical issue remains unresolved. Specifically, the paper does not fully incorporate an appropriate evaluation system for radiation-induced dermatitis (RID). The authors mention, "As radiation oncologists, we are acutely aware of the detrimental effects of ionizing radiation and the ethical considerations involved in exposing animals to such harm." I fully agree with this concern. However, this does not justify keeping the title focused on RID. If the evaluation system used is for contact dermatitis, then the title should reflect this appropriately. The authors have two options moving forward:

    [1] Provide evidence that Black Soybean Extract Cream (BSEC) is useful for RID, supported by a robust set of previous studies (not the authors' own work) demonstrating that DNCB can serve as a model for RID.

    [2] Remove the claims about RID (from Abstract, and Introduction) and significantly revise the manuscript to focus on skin inflammation more broadly. This would still allow for discussion on the potential future use of BSEC for RID. For example, the title could be something like "Exploring Black Soybean Extract Cream for Inflammatory Dermatitis – Toward Radiation Dermatitis Relief." Of course, this is just a suggestion, and the authors are free to choose a title that best fits the revised content. If option [2] is chosen, the relevant statements in the Introduction would also need to be rewritten accordingly.

  2. Related to point 1, in my previous review, I requested a standardized method for evaluating RID. The authors provided a response note defining RID in clinical practice and describing how it is assessed, but they did not incorporate this into the introduction. While the response appears to understand my concern, it does not fully address it. What I was asking for is a standard methodology for evaluating RID in a laboratory setting, specifically using a mouse model. Additionally, the authors should clearly explain in the Introduction what diseases DNCB is being used to model. If they cannot provide this clarification, I would recommend following option [2].